# Intradermal Immunization of Soluble Influenza HA Derived from a Lethal Virus Induces High Magnitude and Breadth of Antibody Responses and Provides Complete Protection In Vivo

**DOI:** 10.3390/vaccines11040780

**Published:** 2023-03-31

**Authors:** Sneha Raj, Preeti Vishwakarma, Shikha Saxena, Varun Kumar, Ritika Khatri, Amit Kumar, Mrityunjay Singh, Surbhi Mishra, Shailendra Asthana, Shubbir Ahmed, Sweety Samal

**Affiliations:** 1Translational Health Science & Technology Institute, NCR Biotech Science Cluster, Faridabad 121001, India; 2Centralized Core Research Facility (CCRF), All India Institute of Medical Sciences, New Delhi 110029, India

**Keywords:** influenza, vaccine, hemagglutinin, intradermal route, virus

## Abstract

Immunogens mimicking the native-like structure of surface-exposed viral antigens are considered promising vaccine candidates. Influenza viruses are important zoonotic respiratory viruses with high pandemic potential. Recombinant soluble hemagglutinin (HA) glycoprotein-based protein subunit vaccines against Influenza have been shown to induce protective efficacy when administered intramuscularly. Here, we have expressed a recombinant soluble trimeric HA protein in Expi 293F cells and purified the protein derived from the Inf A/Guangdong-Maonan/ SWL1536/2019 virus which was found to be highly virulent in the mouse. The trimeric HA protein was found to be in the oligomeric state, highly stable, and the efficacy study in the BALB/c mouse challenge model through intradermal immunization with the prime-boost regimen conferred complete protection against a high lethal dose of homologous and mouse-adapted InfA/PR8 virus challenge. Furthermore, the immunogen induced high hemagglutinin inhibition (HI) titers and showed cross-protection against other Inf A and Inf B subtypes. The results are promising and warrant trimeric HA as a suitable vaccine candidate.

## 1. Introduction

Influenza viruses cause acute respiratory infections with zoonotic potential and are a major pandemic threat globally [1]. Despite the availability of flu vaccines for more than 50 years, flu infections continue to persist and rise causing severe health and socio-economic burden worldwide [2]. The Influenza virus is a segmented, enveloped negative sense RNA virus and belongs to the Orthomyxoviridae family [3]. There are four types of influenza viruses, A, B, C, and D, having a wide range of reservoirs ranging from birds, mammalian hosts, dogs, swine, horses, and bats. Three major types of Influenza A, B, and C infect humans causing mild to severe infections of which the Influenza A virus (IAV) is the dominant type [4]. The available vaccines are made up of the circulating strains of IAV and IBV and provide short-term protection against seasonal viruses [5]. However, the error-prone mechanism of the viral RNA genome, (antigenic drift) and rearrangement of the segmented nature of the genome (antigenic shift) continues to give rise to new strains with varied pathogenesis and virulence [6,7].

The protein subunit-based influenza vaccine development mainly targets the surface glycoproteins hemagglutinin (HA), and neuraminidase (NA), which are the major antigens that induce neutralizing antibodies and provide protection [8]. The IAVs are further classified into subtypes according to the combinations of HA and NA surface proteins. So far, 16 HA (H1-16) and 9 NA (N1-9) subtypes have been isolated from the natural reservoir of aquatic birds [9], and recently two additional HA subtypes H17–H18, and NA10, NA-11 have been isolated from bats [10,11]. HA is present abundantly on the surface of the influenza viruses and initiates the infection by attachment to the host cell sialic acid receptors [12,13]. The receptor-binding further mediates the fusion process, in which the HA protein undergoes multistep conformational changes from prefusion to post-fusion conformation, thus exposing the fusion peptide to initiate the fusion of the virus-host cell membrane [14,15]. Hence, even though the HA protein undergoes the maximum genetic mutation, it is the predominant antigen that elicits potent neutralizing antibodies [16].

The HA glycoprotein mainly forms homologous trimers on the virion surface [17]. Each precursor of HA (HA0) monomer is approximately ~70 KDa and is cleaved by the host cell proteases to HA1 (receptor binding domain or head region) and HA2 (ectodomain or stalk region). The cleavage of HA is an essential prerequisite for the maturation of HA into a functional protein and to initiate the infection [18]. The HA1 domain consists of the globular head domain, which is the main antigenic target for the induction of neutralizing antibodies, however, the globular head domain rapidly mutates [19]. The stalk domain is highly conserved and currently is the major focus for the development of universal vaccine candidates for influenza subtypes [20]. In addition, during post-translation modification in the endoplasmic reticulum (ER) and Golgi apparatus the HA protein undergoes a glycosylation process, thus HA protein is heavily glycosylated which could further influence the virus biology and immune responses [21,22].

The commercially available marketed influenza vaccines are mainly produced by three approaches; (1) inactivated influenza vaccine (IIV), (2) live attenuated influenza vaccine (LAIV), and (3) recombinant HA vaccine [23]. Although every approach has its pros and cons, recombinant subunit HA-based protein production is one of the better choices for pandemic preparedness, as it could be rapidly produced, cost-effective, and could overcome egg allergy [24,25]. However, there are a few disadvantages of a subunit protein-based vaccine which are, a single HA protein might not be a potent immunogen, may lack cross-protection, may require higher and repeated doses, and different adjuvants formulations [26]. Furthermore, in the majority of cases, the subunit protein-based vaccine is administered intra-muscularly, which is not the best choice for children and vulnerable populations (old, immunocompromised, pregnant women) [8]. Another promising vaccine delivery approach is microneedle patches in which the antigen is delivered via Intra dermal route, which will be painless and might enhance the vaccine uptake [27].

In our study, we have generated a recombinant full-length soluble trimeric HA protein derived from the sequence of Influenza A/Guangdong-Maonan/SWL 1536/2019(H1N1) pdm-09-like virus and expressed in mammalian Expi293F expression sys-tem to preserve the native-like structure, termed as HA-T-AGM protein. The 2020–2021 northern hemisphere quadrivalent vaccine consists of four circulating flu strains, an Inf A/Guangdong-Maonan/SWL1536/2019 (H1N1) pdm09-like virus; an Inf A/Hong Kong/2671/2019 (H3N2)-like virus; an Inf B/Washington/02/2019 (B/Victoria lineage)-like virus; and an Inf B/Phuket/3073/2013(B/Yamagata lineage)-like virus (https://www.who.int/publications/m/item/recommended-composition-of-influenza-virus-vaccines-for-use-in-the-2020–2021-northern-hemisphere-influenza-season, accessed on 5 January 2023). Out of these four viruses, the A/Guangdong-Maonan/SWL1536/2019 (H1N1) pdm09-like virus was found to be highly lethal in the BALB/c mouse infection model. We have generated the ectodomain HA protein in its native-like form by stapling a fold-on trimerizing domain at the C-terminal. The recombinant soluble HA-T-AGM protein was found to form trimers and oligomers, highly stable and soluble protein when administered intradermally along with Addavax adjuvant in BALB/c mice, eliciting robust antibody responses. We further evaluated the protective efficacy by challenging the immunized mice with a lethal dose of mouse-adapted Inf A/PR/8/1934 virus. The HA-T-AGM immunized mice showed complete protection against the lethal challenge, elicited potent hemagglutination inhibition (HI) titer, and further showed cross-protection against InfA/H1N1/Cal04, InfA/H3N2/X31, and InfA/H3N2/X79 viruses as measured by HI and microneutralization. The immunized mice sera also showed high HI titer against Inf A/Hong Kong/2671/2019(H3N2) and Inf B/Washington/02/2019 (Victoria lineage). Furthermore, we have also found complete protection against the homologous Inf A/Guangdong-Maonan/SWL1536/2019 virus challenge in immunized BALB/c mice. Our results suggest that the stable higher-order oligomeric HA-T-AGM antigen derived from a lethal virus and expressed in a mammalian expression system is a promising immunogen and could be used for the development of subunit protein vaccine and in the development of self-delivery microneedle patch formulation.

## 2. Materials and Methods

### 2.1. Design, Expression, and Purification of Recombinant HA-T-AGM Protein

The sequence of HA-T-AGM was derived from the HA (aa 18-530) (Inf A/Guangdong-Maonan/SWL1536/2019 (H1N1); GISAID accession#: EPI1542570) and the human codon-optimized gene (Invitrogen) was cloned in pCDNA 3.1 expression vector. The HA gene was fused to the CD5 signal peptide sequence at the N-terminal for extracellular expression and attached with the Foldon domain at the C-terminal followed by an avi-tag, a TEV cleavage site, and 6x-His tag for ease of purification.

For protein production, Expi293F cells were maintained in suspension cultures in Expi Expression Medium (Thermo Fisher Scientific, Waltham, MA, USA) at 37 °C, 8% CO_2_, and 80% humidity on a shaker incubator set to 110 rpm, cell density was maintained at 0.5–8 × 10^6^ cells/mL in polycarbonate vented Erlenmeyer flasks (Corning, NY, USA) containing a medium volume of 1/3 of the total volume of the flask. Cells were transfected using the Expifectamine 293 Transfection Kit (Gibco^TM^) according to the manufacturer’s instructions. Target plasmid DNA was diluted using Opti-MEM in 5% of the final culture volume, while in a separate conical tube, Expifectamine was diluted in 5% of the final culture volume to achieve a final culture concentration of 1 μg DNA/mL. Diluted mixtures were incubated for 5 min at room temperature (RT). DNA mixture was added to the Expifectamine mixture and incubated at RT for 30 min before addition to the cells. After 18–20 h post-transfection, 0.5% of Enhancer 1 and 5% of Enhancer 2 to the final culture volumes were added. Expressed protein in the culture supernatant was harvested 5–6 days post-transfection or when viability reached <60%.

The harvested supernatant was directly loaded onto the PBS equilibrated nickel-nitrilotriacetic acid agarose, Ni-NTA resin (Qiagen, Germany), and protein was allowed to bind. Flowthrough was collected thereafter with an increasing gradient of imidazole; unwanted loosely bound proteins were removed first with 30mM imidazole and thereafter small fractions of protein were collected and confirmed with Bradford’s reagent with final elution from 500 mM imidazole. The eluted protein was dialyzed against PBS to get rid of the suspended imidazole fully. For size exclusion chromatography (SEC), a HiLoad Superdex 200 Increase 10/300 GL column (GE Healthcare, Chicago, IL, USA) was used, and PBS was used as the mobile phase with a flow rate of 0.5 mL/min. The purified protein aliquots were snap-frozen in liquid nitrogen and stored at −80 °C until further use. Protein purity and oligomeric status were confirmed in SDS-PAGE and gradient 4–15% Native-PAGE.

### 2.2. SDS-PAGE, Native PAGE, and Western Blotting

Sample proteins were prepared by boiling for 10 min with SDS and β-Mercaptoethanol consisting of loading dye and resolved on reducing 12% SDS-PAGE, followed by visualization with Coomassie brilliant blue staining. For Native-PAGE analysis, the proteins were separated by 4–15% Native-PAGE gels (Mini-PROTEAN TGX^TM^, Bio-Rad, Hercules, CA, USA) using Native-PAGE sample preparation buffer (Invitrogen, Waltham, MA, USA) and Bis-Tris running buffer, followed by the same protocols as above.

For further characterization with western blotting, the separated proteins on SDS-PAGE were transferred to nitrocellulose membrane by electroblotting (Biorad, USA) for 1 h at room temperature at 100 V. The membrane was then blocked with blocking buffer (BSA 1%) at room temperature for 1 h. The membrane was probed with anti-mouse sera (1:500) and separately with monoclonal commercially available antibody (1:1000, IRR FR572),. The membrane was washed three times with washing buffer [0.5% (*v*/*v*) Tween 20, 1 × PBS], each for 10 min at room temperature and then probed with anti-mouse IgG HRP conjugated antibody (1:2000) for another 1 h before washing. The blots were developed using chemiluminescence reagents (luminol and peroxidases from G Biosciences, St. Louis, MO, USA).

### 2.3. Vaccination and Challenge Study in BALB/c Mice

BALB/c mice of 6–8 weeks of age were used for this study, which were inbred at the THSTI small animal facility. All experiments were conducted to minimize animal suffering, and carried out following the principles of humanity described in the relevant Guidelines of the CPCSEA, the protocol was approved by the Institutional Animal Ethics Committee (IAEC Approval number: IAEC/THSTI/147). In the preliminary experiment, we assessed the pathogenicity of the Inf A/Guangdong-Maonan/SWL1536/2019 virus in BALB/c mouse. Briefly, the virus was grown in MDCK cell line and the virus titer was caluclated 6–8 weeks old naïve BALB/c mice (*n* = 5) were challenged intranasally with 10^5^ TCID_50_/mL of viruses, and animals were monitored for their body weight and survival post-challenge. For the immunization study, the animals were randomly divided into three groups (*n* = 5) each for the soluble immunogen and virus control group and PBS control). The animals were gender-matched and randomly divided into experimental groups such that the average weight of the animals in each group was +/− 10% of the body weight of each group and of the individual animals. Mice were injected with 30µg of antigen (HA-T-AGM) with Addavax adjuvant in a 1:1 ratio. The immunization was carried out with a single prime-boost strategy on day 0 and day 21 respectively. For i.d. delivery, mice were injected on the lower dorsal surface using a 1-mm-long, 34-gauge (Ga) stainless steel microneedle (15, 34, 36) fitted to a 1-mL syringe (BD, Franklin Lakes, NJ, USA) and inserted perpendicularly into the skin to control delivery depth [28]. The total volume injected per mouse was 100 µL. For serum extraction, mice blood samples were collected and centrifuged at 4000 rpm to separate the sera. Collected sera were heat inactivated at 56 °C for 1 h for complement deactivation before they were used for ELISA and HI experiments. The vaccinated mice were then challenged on the 42nd post-immunization day, intranasally (i.n.) with 10 MLD_50_ of the highly virulent mouse-adapted Inf A/Puerto Rico/8/34 (PR8) virus. Similarly, in another set of experiment recombinant HA-immunized mice, *n* = 5 (30 µg intradermally given twice (0 prime and 21 days boost) were challenged on 42nd day of immunization with 10^5^ TCID_50_/mL of homologous virus Inf A/Guangdong-Maonan/SWL1536/2019 (H1N1) virus through intranasal administration. Subsequent body weight changes, clinical score, and total survival were followed up for the next 14 days thereafter.

### 2.4. Antigen Binding ELISA

An indirect ELISA was performed for testing the immune response against HA-T-AGM as described earlier [29] with minor modifications. Briefly, Maxisorp^TM^ ELISA plates (Nunc, Roskilde, Denmark) were coated either with 1 µg of HA-T-AGM, or commercially available known antigen (IRR FR695, Influenza A H1) using carbonate–bicarbonate buffer (pH 9.6) and phosphate buffer (pH 7.4) separately and incubated overnight at 4 °C. The plates were washed twice with washing buffer (0.1 M PBS containing 0.05% Tween 20; PBST) and blocked with blocking buffer (5% skimmed milk in PBST) at 37 °C for 1 h. The serum samples were added at two-fold dilutions from 1/200 to 1/25,600 in sample buffer (0.5% skimmed milk) and incubated at 37 °C for an hour. After incubation, the plates were washed four times with PBST and anti-mouse horseradish peroxidase (HRP) conjugate antibody (Jackson Immuno Research) was added to all the wells at a dilution of 1:2000 and incubated for 1 h. After washing four times with wash buffer, the plates were developed with TMB (3,3′,5,5′-Tetramethylbenzidine), and the reaction was stopped after 10 min with 1 M H_2_SO_4_. The absorbance was read at 450 nm.

### 2.5. Hemagglutination Inhibition (HI) Assay

Hemagglutination inhibition assays were performed as described earlier [30] with minor modifications. Briefly, the serum samples were pre-treated with receptor-destroying enzyme (RDE) (Denka Seiken, San Jose, CA, USA) in a proportion of 1:3 volume of sera to RDE for 16 h at 37 °C before inactivation at 56 °C for 30 min. The serum samples were diluted serially from 1/4 to 1/1024 with PBS. An equal volume (50 µL) of serum and Influenza virus of different subtypes (4 HA units) were mixed and incubated for 30 min at room temperature. After incubation, an equal volume of 1% (*v*/*v*) freshly prepared chicken red blood cells in PBS was added and the plates were observed after 45 min. The results were interpreted by observing button formation for different dilutions without tilting the plate as considered appropriate by Wilson et al. [31]. The hemagglutination inhibition titer was calculated as the reciprocal of the highest serum dilution showing clear button formation and lattice formation as hemagglutination.

### 2.6. Circular Dichroism (CD) Assay

The stability studies were carried out at ambient temperature (25 °C) in phosphate buffer using a Jasco J-815 spectropolarimeter from 195 to 250 nm wavelength, using a 1-mm path length quartz cell. Data were collected at a rate of 100 nm/min for each protein, averaging 3–5 scans as necessary. The concentration of protein used ranged between 5 and 10 mM. The wavelength dependence of molar ellipticity was monitored at 24 °C as the average of five scans, using a 5-s integration time with 0.5-nm bandwidth at 1.0-nm wavelength increments. For collecting CD spectral data for thermal melt (range 25 °C to 90 °C), the Peltier-controlled cuvette holder attachment of the spectropolarimeter was used, with 8-mm spacers for heat transfer to 2-mm cuvettes. Background spectra from the buffer were electronically subtracted and for each spectrum, mean residual ellipticity was calculated and plotted. The fractions of the secondary structure elements were calculated by minimizing the difference between the theoretical and experimental curves by varying the impacts of the α-helixes, β-sheets, turns, and nonstructured forms. Theoretical values at every wavelength were the linear combination of the basis spectra of every type of secondary structure.

### 2.7. Dynamic Light Scattering (DLS)

The synthesized HA-T-AGM protein was further characterized for its size, zeta potential, and polydispersity index through Malvern Zetasizer UK (Nano ZS). Briefly, 1–5 mg/mL of HA-T-AGM protein was prepared in PBS (pH 7.0) and sonicated for 4–5 min. The prepared samples were then monitored for size, zeta potential, and polydispersity index using a Nano ZS analyzer in a triplicate manner, and the observations were represented as Mean ± SD.

### 2.8. Immunofluorescence Microscopy

MDCK-London cells (0.2 million/well) were seeded in a 12-well plate. The next day, cells were infected with different influenza variants: PR8 (Inf A/Puerto Rico/8/1934(H1N1), Guangdong (Inf A/Guangdong-Maonan/SWL1536/2019 (H1N1)), X-31 (Inf A/Hong Kong/X31(H3N2)), X-79 (Inf A/Philippines/2/82 (H3N2) and Cal09 (Inf A/California/04/2009 (H1N1)) at an m.o.i. of 0.5. After 1 h of adsorption, the plate was washed once with Advanced Modified Eagle Medium, and virus growth media (VGM) was added to the infected wells. Cells were fixed after 24 h of infection in 4% paraformaldehyde for 15 min, then penetrated with 0.1% triton-X in PBS for 10 min. Non-specific binding was blocked using 3% goat serum in PBS for 1 h at room temperature. Cells were then incubated overnight at 4 °C with the Guangdong anti-HA polyclonal sera (1:200). Next day, cells were washed three times with PBS and incubated for 1 h at room temperature with Alexa488-labeled rabbit anti-mouse IgG (1:1000). Three washes were given and the cell nuclei were counterstained with DAPI (D9542, Sigma-Aldrich, Burlington, MA, USA) for another 10 min at room temperature. The expression of proteins was observed by fluorescence microscope (IX-71, Olympus).

### 2.9. Molecular Modelling and Protein-Protein Docking

To generate the 3D structures of identified Influenza A AGM serotype soluble protein (HA-T-AGM), the crystal structure 4LXV (chains A and B, both) was selected. There were multiple templates identified through BLAST (search against PDB) that were closer to AGM serotype, though, we have picked PDB-id 4LXV (Crystal Structure of the Hemagglutinin from an H1N1pdm A/Washington/5/2011 virus taken from www.rcsb.org) considering it as wild type. The multi-template-based molecular modeling was carried out using Modeller software to generate the 3D structure of HA-T-AGM [32,33].

The modeled structure was used for protein-protein docking studies. The PIPER module of Schrodinger suits was used to identify the most likely poses of the complex between trimer HA and monoclonal antibody (3SDY). To find the most likely binding pose. The top three highly enriched conformations and docking energy was picked for pose analysis.

### 2.10. Ethics Statement

The animal studies were carried out in strict accordance with the recommendations in the Guide for the Care and Use of Laboratory Animals of THSTI, Faridabad, India. The protocol was approved by the Committee on the Ethics of Animal Experiments and all experiments related to the influenza viruses were performed in an approved biosafety level 2 (BSL-2) laboratory or ABSL-2 small animal facility.

### 2.11. Statistics

Statistical Analysis Values were presented as mean with SE. Statistical significance among different groups for analysis of titer was calculated by one-way ANOVA using Graph Pad prism 8.0 statistical software. The data sets obtained from different groups in the challenge study were subjected to a test for normal distribution of the samples by the Shapiro-Wilk normality test; statistical analysis was then performed using Student’s *t*-test or Dunnett’s multiple comparison tests where necessary. A *p*-value less than 0.05 was considered significant.

## 3. Results

### 3.1. Generation of Recombinant Soluble Hemagglutinin Immunogens in Expi293F Cells to Preserve the Native-like Structure

We designed and generated a full-length sequence, consisting of both head and stalk domains, of HA from Inf A/Guangdong-Maonan/SWL1536/2019 (H1N1) pdm09-like virus. The Inf A/Guangdong-Maonan/SWL1536/2019 (H1N1) pdm09-like viruses are the circulating strains from the 2020–2021 northern hemisphere influenza season and belong to the subclade 6B.1A5A [34]. The virus was shown to be 90% pathogenic in mice [35]; in our study, we also found that among the NH2020 vaccine virus composition, the A/Guangdong-Maonan/SWL1536/2019 virus is highly virulent in BALB/c mice and intranasal inoculation (50 µL) of 10^5^ TCID_50_ /mL of viruses resulted in 100% mortality by 8th-day post-infection (dpi) (Appendix A). We predicted the 3-D homotrimer structure representing the HA molecule (Figure 1A). The HA structure is a mixture of both α-helices and β-sheets intermittently [36]. In this case, the transmembrane region of HA from Inf A/Guangdong-Maonan/SWL1536/2019 (H1N1) virus is predominantly occupied with α-helical secondary structure (shown in purple, Figure 1A) owing to the most preferred conformation to accommodate hydrophobic residues whereas distal ends with receptor binding domain (shown in green, Figure 1A) are abundant in β-sheets giving elongated and extended surface areas for receptor interactions. Interestingly, the HA-T-AGM protein sequence also shows 97% sequence homology with Influenza A/Michigan/45/2015 (H1) protein (Data not shown). The A/Michigan/45/2015 strain did not lead to a pandemic but its occurrence in humans and pigs suggests it could be a future threat. Hence, we sought to make the soluble recombinant HA protein against the Inf A/Guangdong-Maonan/SWL1536/2019 virus. The HA amino acid sequence 18–530 was used for the generation of HA protein and linked to the fold on trimerization domain at the C-terminus for trimer formation and to preserve the native-like trimeric conformation of HA [37,38,39]. The sequence was further linked with an avidin (avi) tag, cleavable tobacco etch virus (TEV) protease site, and a His6 tag at the C terminus for ease of purification (Figure 1B). The engineered construct was expressed in Expi293F mammalian expression cells for proper glycosylation which would facilitate mimicking the native-like structure. The soluble proteins (HA-T-AGM) were purified using affinity purification by passing over Ni-NTA immobilized metal ion column and characterized further. We performed three batches of small laboratory-scale purification and the yield of the soluble HA protein was ~6–8 mg/L. We analyzed the expression of the purified protein on a 12% SDS-PAGE and found a single band migrating around ~75 KDa (Figure 1C), [40]. The results confirm the expression of trimeric HA, although the yield was moderate.

### 3.2. Comprehensive Antigenic Characterization of HA-T-AGM Soluble Protein Reveals Highly Stable Antigen

Next, we assessed the soluble protein migration on SDS-PAGE under reduced and non-reduced conditions. No difference in migration pattern was seen under reduced or non-reduced conditions indicating no possible aggregate formation (Figure 2A). The protein was expressed in Expi293F cells, followed by size exclusion chromatography using a Superdex increase 200 10/30 column. An elution peak at volume ~11 mL corresponding to molecular weight ~240 KDa was observed (Figure 2B). Although some minor peaks were observed in the gel filtration chromatogram that may correspond to other oligomeric forms, however, the trimeric population showed to be >90% population. We further test the trimeric nature of the HA-T-AGM protein using the blue-native page electrophoresis. Under native conditions, the protein was seen in its various oligomeric states ranging from molecular weights from 480 KDa-720KDa (Figure 2C). To check the antigenicity of the recombinantly produced soluble HA proteins, we performed western blot analysis using two different -anti-HA-antibodies as a primary probe. Polyclonal homologous anti-HA-T-AGM mouse sera raised in-house and a commercially available monoclonal antibody directed against H1 of type A pdm09 Influenza virus (IRR FR 572); both polyclonal and monoclonal HA-specific antibodies recognized the soluble HA-T-AGM protein (Figure 2D) (Appendix A for full blot images) Next, the binding of the HA-T-AGM soluble protein was further checked by ELISA using the above antibodies. Results indicated high binding by the soluble HA-T-AGM protein to both the homologous sera and heterologous mAb (Figure 2E). We checked the binding capacity of the soluble HA-T-AGM at two different pH buffers, and we found that pH 7.4 gave a better ELISA readout than pH 9.6 (Appendix A). This was confirmed by several previous reports suggesting that pre-treatment of antigen at lower a pH (<5.0) shows better ELISA readouts owing to the natural physiological events of Influenza replication biology. This involves occurring of conformational changes in the HA molecule to expose the HA1 region for better antibody binding [41]. Out of the two chosen buffer systems for in-house developed ELISA, we found pH 7.4 functioning better than pH 9.6 buffer.

### 3.3. Stability and Biophysical Characterization of Recombinant Soluble HA-T-AGM Protein

We evaluated the stability of the HA-T-AGM protein by incubating equal concentrations of HA-T-AGM protein aliquots at 4 °C and 37 °C and their stability was checked on 12% SDS-PAGE w.r.t different time points (Figure 3A). The HA-T-AGM protein showed better stability when stored at 4 °C and even after 21 days of storage, no degradation of the protein was noticed and a compact single band of ~75 kDa was seen on SDS PAGE. Whereas upon storage at 37 °C, the protein was found to be degrading after 72 h of incubation. Next, we assessed the secondary structure and stability of the HA-T-AGM protein by circular dichroism spectra analysis. Far-UV CD spectra of the two proteins were collected between 280 and 195 nm. The single negative peak at ~217 nm of HA-T-AGM suggested that the protein adapts mostly to an antiparallel β-sheets structure for their secondary orientation. However, this did not overlap completely with the standard graph indicating there are other secondary structural forms also present such as α-helical structures in the overall structure (Figure 3Bi). To test whether the stability is dependent on temperature, the sample was heated from 25 °C to 90 °C gradually, and a fall in negative ellipticity was measured. Results indicated that ellipticity remained almost unchanged until temperature 54 °C and thereafter, gradual decline in the negative values of ellipticity suggesting that the protein has stability until this temperature range and no major structural loss till 54 °C (Figure 3Bii). These results showed that the soluble protein is well-folded and functionally stable over a good range of temperatures. The size distribution profiling of HA-T-AGM was determined using dynamic light scattering (DLS) at 25 °C with light scattering at 90° for individual measurement and found to be 446.3 ± 3.05 nm with a PDI value of 0.46. The estimated PDI value supports the narrow size distribution of the synthesized HA-T-AGM in suspension. The observed larger size of HA-T-AGM could be explained by the presence of a multimeric form (HA trimer) of the synthesized protein. The apparent zeta potential of HA-T-AGM was analyzed in the PBS buffer (pH 7.0). The ionic strength and pH of the diluents have a great impact on the magnitude of the zeta potential. The synthesized proteins were negatively charged and zeta potential was depicted to be −40.86 ± 3.02 mV. The observed values for zeta potential indicate the high colloidal stability of the synthesized protein as high values for zeta potential lead to decreased attraction among the particles and therefore show less agglomeration. The observed values for zeta potential and size distribution profiling favor the synthesized HA-T-AGM protein’s high stability and narrow size distribution (Figure 3C).

### 3.4. Intradermal Administration of HA-T-AGM Soluble Antigens in BALB/c Mice Elicited Robust Humoral Responses and Provides Complete Protection against High-Dose Lethal Challenge

BALB/c mice were used for immunization with recombinant soluble protein HA-T-AGM (30 µg) along with a squalene-based oil-in-water nano-emulsion based on the formulation of MF59^®^ known as Addavax^TM^. The antigen-to-adjuvant ratio of 1:1 was used for intradermal immunization. We followed one prime-boost strategy for the immunization with a gap of 21 days and challenged the immunized mice with a high lethal dose of mouse-adapted InfA/PR8 virus (10 MLD_50_) on day 42 post-immunization (Figure 4A). The serum samples were collected post-14 days of each dosing and used for serological assays to assess the induction of serum-specific whole IgG (H+L). A significant rise in the whole IgG antibody titer was seen in the sera from prime to single boost in the immunized mice against the homologous HA-T-AGM proteins (Figure 4B). Further assessment of the type of immune responses generated was determined in the immunized sera by measuring IgG1, IgG2a, and IgG3. Th_1_:Th_2_ response index was calculated using the endpoint titer values in the formula [(IgG2a + IgG3)/2]/[IgG1] and the determined value was 0.63 indicating a skewness of immune response towards Th_2_ polarization. Anti-HA-T-AGM mouse sera were also tested for binding with heterologous proteins of Influenza viruses. The two commercially available proteins from Influenza A H1N1 and H3N2 strains, namely FR180 and FR401 were tested. Endpoint titers for homologous protein binding were found to reach a maximum of up to 10^6.5^, whereas heterologous binding was only less by around 1 log value in the case of the H1N1 strain’s protein and 2 log value in the case of the H3N2 strain’s protein binding (Appendix A). The binding of immunized sera with different subtypes derived proteins suggests a wide breadth of antigenicity.

Next, we evaluated the protective efficacy in the HA-T-AGM mice by challenging the immunized mice at 42nd post-immunization, intranasally (i.n.) with 10× mouse 50% lethal dose (10 MLD_50_) of the highly virulent mouse-adapted Inf A/Puerto Rico/8/34 (PR8) virus. The mice were monitored daily up to 14 days post-challenge for the development of clinical signs. In the virus control group, all the mice developed clinical signs such as dullness, shivering, piloerection, reduction in food intake, and hunched back. From the 3rd day post-infection, virus-challenged mice showed a loss in body weight, and by the 10th-day post-challenge, all the mice were dead (Figure 5A–C). However, we observed one prime-one boost of HA-T-AGM immunogens was able to protect 100% of the immunized mice against the lethal challenge with 0% mortality as compared to the control group that succumbed fully by the 10th day of infection. We further performed the challenge study with homologous virus Inf A/Guangdong-Maonan/SWL1536/2019. The mice (*n* = 5) were immunized with HA-T-AGM similarly as above, and on the 42nd day post-immunization mice were challenged with 10^5^ TCID_50_ /mL of Inf A/Guangdong-Maonan/SWL1536/2019 intra nasally. The mice were monitored daily for body weight, survival, and clinical signs (Appendix A. The immunized mice showed complete protection against the homologous virus challenge.

### 3.5. Qualitative and Quantitative Evaluation of Magnitude and Breadth of Cross-Protection as Exhibited by Immunized Mouse Sera

We used the gold standard test for quantifying anti-HA antibodies by using a Hemagglutination Inhibition assay which shows the correlation of protection. Our data shows mouse sera protected against tested homologous virus i.e., Influenza A/Guangdong Maonan/2019 strain with a high HI titer of >16,384 (Figure 6A and Appendix A). For other heterologous viruses of the whole vaccine composition recommendation by WHO for NH2020 (Appendix A), the HI titer was 256 for Inf A/Hong Kong/2671/2019(H3N2) and 2048 for Inf B/Washington/02/2019 (Victoria lineage); additionally, subtypes such as H1N1 Cal/04 strain, H3N2 X-31, and H3N2 X-79, the HI titer for these three subtypes was determined to be around 1024 (Figure 6B and Appendix A). The microneutralization ability of the mouse sera was also tested along with immunofluorescence assay (IFA) and protection against homologous and heterologous virus subtypes was shown (Figure 6C,D). The highest microneutralization was shown against the homologous virus Influenza A/Guangdong Maonan// SWL1536/2019, followed by heterologous Influenza A/Philippines/2/82 (H3N2) X-79 virus. Immunofluorescence was also conducted independently and readouts were measured separately as fluorescence intensity (Appendix A). Overall results from this study show HA-T-AGM immunogen as a potent vaccine candidate.

### 3.6. In Silico Structural Analysis of HA-T-AGM Protein

To understand the HA-T-AGM structure we conducted the computational simulation as described in experimental procedures. The crystal structure of the hemagglutinin from the H1N1pdm A/Washington/5/2011 virus was taken as a template for the generation of a 3D structure. We performed the sequence alignment between the target (HA-T-AGM serotype) and 4LVX (Crystal Structure of the Hemagglutinin from an H1N1pdmA/Washington/5/2011 virus taken from www.rcsb.org. The sequences are highly conserved and the changes were observed at N- and C-terminals of HA protein mainly (Appendix A). Furthermore, there are some amino acid changes that are highlighted in the alignment file. The 3D structure of HA-T-AGM was generated using Modeller. 200 models were generated and the topmost model which has shown the lowest potential energy (in kcal/mol) was picked to compare the structural changes with crystal structure using both chains A and B. The model was optimized using the OPLS3 force field and optimization was carried out using Schrodinger software. The optimized model was superimposed against chains A and B of 4LXV and the RMSD was observed at 0.62 Å, indicating that the model is structurally similar to crystal structure. Furthermore, we noticed the amino acid changes concerning crystal, majorly occurring at positions 186-to-188 and 207-to-222. Some individual changes were also observed in amino acids that are R98S, N108S, N121D, D153N, D284N, E307K, V319I, and K523E. The optimized monomer models were used to establish the trimer model which includes all three protomers (Figure 7A).

We next conducted the molecular docking studies of 3D generated HA-T-AGM with trimer-specific CR8020 mAb reported PDB structure as 3SDY (Figure 7B). The mAb CR8020 was optimized using the protein preparation wizard of Schrodinger. The piper was used for the docking between HA-trimer and mAb. Based on docking energy the top three poses were filtered to explore their binding. The structures with docking energies of −317, −303, and −282 kcal/mol were shortlisted. The most energetically favorable pose was localized at the stalk region of HA-T-AGM; however, the other two poses belong to the tail and head regions, respectively.

## 4. Discussion

Despite the extensive antigenic variability of Influenza HA structural protein, this is the major antigen that confers protection against Influenza viruses [42]. Current licensed seasonal vaccines are also focused on the protective ability of the HA glycoprotein to elicit potent neutralizing responses, although these vaccines confer only strain-specific protection [43]. An ideal vaccine for Influenza should have a greater breadth of protection and should be suitable for all age groups. Novel and promising approaches are currently under evaluation for the development of next-generation universal vaccines [44]. However, the old paradigm of using recombinant subunit HA-based vaccine against the seasonal or pandemic virus strain remains a lucrative approach [45,46]. The subunit vaccine eliminates the concerns of some of the whole virus-based vaccines such as egg-based allergy, pre-existing immunity, and improper inactivation [47]. For Influenza viruses, several HA-based subunit vaccines have been evaluated which provide various levels of protection and breadth [26,48,49]. It has been well-documented that the HA globular head-based vaccines are more protective than stem-based vaccines alone [50].

In the present study, we have designed, expressed, and purified Influenza A HA trimeric recombinant protein from a highly lethal strain of NH2020 circulating strain Influenza A/Guangdong-Maonan/SWL1536/2019(H1N1) pdm09-like virus (Figure 1). The ancestral 2009 pandemic virus strain Inf A/California/7/2009 was non-pathogenic in mice and does not produce any disease symptoms when inoculated intranasally (data not shown); however, Influenza A/Guangdong-Maonan/SWL1536/2019(H1N1) causes high mortality in mice when inoculated intranasally and also the HA is found to be highly stable [35]. In our study also the virus was found to be highly lethal in mice ( Appendix A). Hence, the overall goal was to design the HA antigen by understanding the structure-function relationship, where we hypothesize that soluble native-like HA protein derived from a highly lethal virus and expressed in a suitable expression system might confer better antigenicity and immunogenicity through a yin-yang mechanism.

The yield of the trimeric HA-T-AGM protein was ~6–8 mg/L in the laboratory conditions in the Expi293F mammalian cells, which could be scalable and provides a rapid methodology to adapt to the pandemic threat. The protein forms higher-order oligomers and a highly stable structure and is further recognized by both homologous and heterologous antibodies (Figure 2 and Figure 3).

The subunit protein-based vaccines are generally administered through intramuscular or subcutaneous routes due to the safety and rapid absorption to the circulation [51], however, this route of vaccination is often painful and also difficult to administer in needle phobia patients. In contrast, intradermal administration provides an ideal site that is rich in immune cells, mainly the antigen-presenting cells, and thus might allow longer duration, elicitation of antigen-specific strong memory response by the quick orientation of antigens through draining lymph nodes [28,52]. The assessment of the vaccine efficacy administrated through the intradermal route also allows the development of a microneedle patch delivery system [53].

Thus, we further tested the efficacy of the HA-T-AGM immunogen along with Addavax adjuvant in BALB/c mice when administered intradermally, using one prime-one boost approach. The HA-T-AGM immunization elicited robust and potent HA-T-AGM specific antibody responses even after single dose immunization and after boosting, the peak IgG responses were substantially increased by 2 log (Figure 4B). The immunized sera were also shown to induce a breadth of antibody responses, and shown to bind with HA proteins of H1N1 and H3N2 viruses (Appendix A). A further challenge of immunized mice with a high lethal dose of both homologous Influenza A/Guangdong-Maonan/SWL 1536/2019(H1N1)viruse and mouse-adapted Inf A/Puerto Rico/8/34 (PR8) showed complete protection in immunized mice. Furthermore, the immunized mice induced high HI titer as measured in vitro against the homologous virus (Figure 5). This is an interesting finding that showed the HA-T-AGM trimeric immunogen elicits high neutralizing antibody responses not only against the homologous H1N1 virus but there was also significant induction of neutralizing antibody responses against the H3N2 viruses and also against the Influenza B virus (Figure 6 and Appendix A). This virus is recommended as a part of the northern hemisphere egg-based flu vaccine strain in the year 2020–2021, a part of the Influenza trivalent vaccine (https://www.medicines.org.uk/emc/product/10444/smpc/print#gref, accessed on 21 December 2022), and FluBlok (https://www.sanofiflu.com/flublok-quadrivalent-influenza-vaccine/, accessed on 21 December 2022); however, in this trivalent vaccine, the recombinant protein was made in the baculovirus expression system and this is administered intramuscularly. The glycosylation patterns of the protein expressed in the baculovirus and mammalian expression systems are different. Our results indicate, expressing the trimeric HA in the mammalian expression system might be allowing proper folding and required glycosylation thus allowing the display of neutralizing epitopes in the expressed HA protein which might be conserved across the strains, this may have resulted in the induction of broadly reactive antibody responses.

Additionally, molecular docking studies with trimer-specific monoclonal antibody group 2 CR8020 mAb showed binding to the stem of HA-T-AGM 3D structure (Figure 7). The mAb CR8020 is a broadly neutralizing HA-stem-directed monoclonal antibody that neutralizes the group 2 Influenza viruses which includes H3N2 and H7N7 viruses [54,55] and this antibody targets the highly conserved stem region and is hence shown to be effective in IAV viruses. The immunized sera were able to neutralize H3N2 viruses (X31 and X79). Hence, it is likely that the HA-T-AGM protein is capable of inducing broadly stem-directed neutralizing antibodies.

There were two major limitations in this study. Firstly, we showed protection in the in vivo challenge experiments using the homologous and mouse-adapted strains which were conducted in a low number of mice per group; although early studies have used a similar number of mice in studying the efficacy of novel vaccine candidates [56,57]. However, additional in vivo studies to measure whether the immunized sera could provide prophylactic and therapeutic protection against different Influenza viruses challenge would strongly support these preliminary findings. Nonetheless, the present study directly compares the protection against the high titer homologous and heterologous challenged viruses and there were consistent findings that demonstrated the protection in the immunized mice. Next, more biochemical and biophysical studies are required to further validate the binding of CR8020 mAb to the HA-T-AGM protein to corroborate the molecular docking studies or other broadly neutralizing antibodies binding at different sites on the HA-T-AGM. It will be interesting to find out whether HA-T-AGM antigen could elicit broadly neutralizing antibodies and identify the target domain which would further facilitate the structure-based designing of novel vaccine candidates.

Nevertheless, our study suggests HA-T-AGM is an effective subunit protein-based vaccine that showed potential breadth, which is an essential requisite of the current Influenza vaccine development.

## 5. Conclusions

Altogether, our data demonstrates that native-like HA trimeric antigen derived from a lethal strain (Influenza A/Guangdong-Maonan/SWL 1536/2019(H1N1)) is a novel subunit recombinant vaccine candidate capable of inducing humoral and protective neutralizing antibodies against homologous and heterologous strains when immunized through the intradermal route. Although, in this study, we could not establish the linkage between lethality and higher antigenicity and breadth of the immune response, however, in the future, the development of soluble trimeric HA from virulent strains could further substantiate this phenomenon. The currently used Influenza A/Guangdong-Maonan/SWL 1536/2019(H1N1) HA sequence also has high similarity with two known highly pathogenic H1N1 strains (Influenza A/Hawaii/2570/2019/H1N1 and H1N1pdm A/Washington/5/2011). Hence, it is likely that the trimeric HA-T-AGM protein will be able to protect against heterologous lethal viruses. Our results further suggest the usage of a protein expression platform might play an important role in the development of suitable protein-based vaccine candidates for Influenza. Here, the usage of the mammalian expression system might be resulting in proper folding and display of glycosylation pattern that is resulting in suitable HA conformation to elicit protecting antibody responses. Additionally, the most preferred route of administration of the protein vaccine is the intramuscular route, in this study the intradermal administration of the HA-T-AGM induced potent and broad antibody responses. This soluble recombinant HA-T-AGM protein could also be a potential immunogen for the development of microneedle patches form of vaccine that would offer the advantage of self-administration with ease and less pain.

## Figures and Tables

**Figure 1 vaccines-11-00780-f001:**
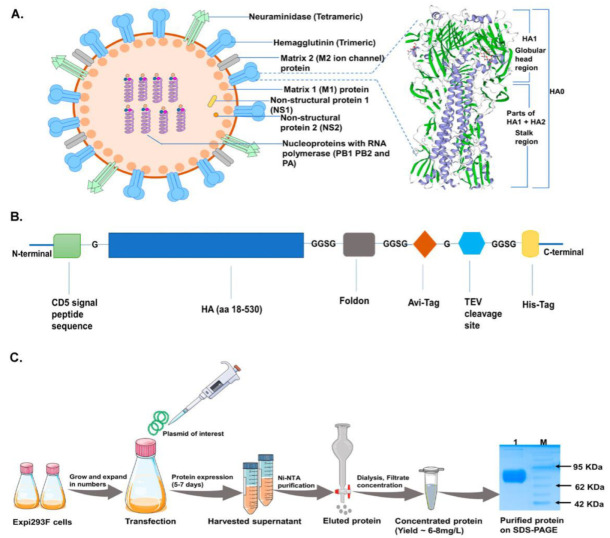
Construct design and protein production scheme in mammalian cells. (**A**): Typical Influenza virion structure along with a magnified view of HA-T-AGM 3-D structure (homo-trimer) as generated by the Swiss-model ExPASy online tool based on secondary structure features (purple strands show α-helices and green highlight are β-sheets). (**B**): Schematic representation of expression cassette used in mammalian pcDNA3.1 expression vector (**C**): Schematic diagram showing expression and purification of HA-T-AGM protein in Expi293F cells i.e plasmid transfection and protein production and purification stages with final eluted fraction, dialyzed and run on 12% resolving gel.

**Figure 2 vaccines-11-00780-f002:**
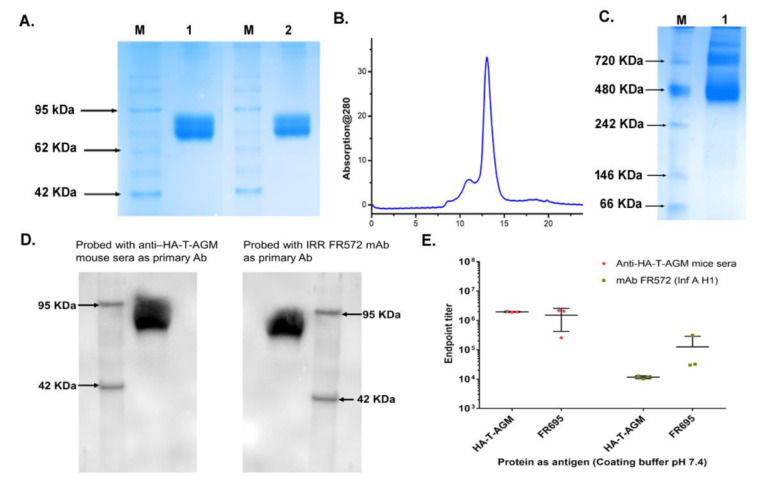
Antigenic characterization of HA-T-AGM protein. (**A**): SDS-PAGE results showing purified protein under reduced and non-reduced conditions; *Lane 1* with reduced HA-T-AGM and *Lane 2* with Non-reduced HA-T-AGM; M is a pre-stained marker (**B**): Elution profile after purification with Superdex 200 Increase 10/300 column showing a peak at elution volume ~11 mL. (**C**): Native-PAGE showing oligomerization states of HA-T-AGM soluble protein (**D**): Immunoblotting of the recombinant HA-T-AGM protein onto PVDF membrane, probing with polyclonal mouse sera (1:500) as primary antibody and another blot with monoclonal antibody IRR FR572 (1:1000) followed by secondary IgG-HRP tagged anti-mouse antibody (1:2000) to develop the blot with help of chemiluminescent substrates (**E**): ELISA results showing the endpoint titers of HA-T-AGM protein and commercial Influenza HA1 from Influenza A/California/07/2009 (H1N1) pdm09 [FR695]. The binding efficiency of both proteins is tested against polyclonal mouse sera and mouse mAb FR572 raised against Influenza Type A (H1) pdm09.

**Figure 3 vaccines-11-00780-f003:**
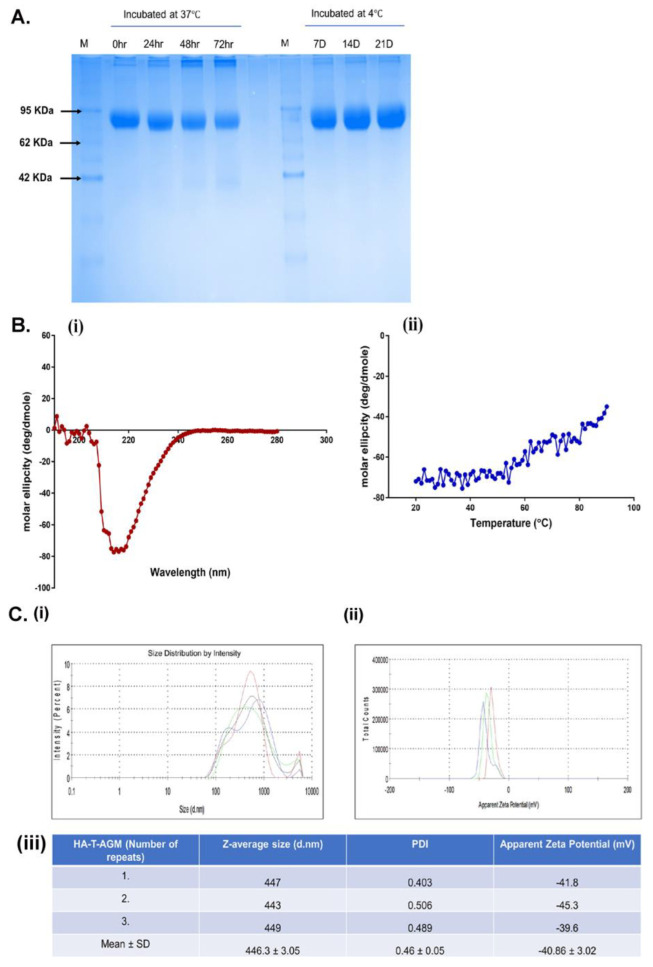
Biochemical and biophysical characterization of HA-T-AGM. (**A**): Thermostability profile of the protein after incubation at temperature for hours at 37 °C and days at 4 °C. (**B**) (**i**,**ii**): CD spectroscopy of HA-T-AGM protein. (**i**) Far-UV CD spectra in the wavelength range of 195–280 nm. (**ii**) Thermal unfolding of HA-T-AGM as monitored from 25 to 90 °C. (**C**): Biophysical characterization of the synthesized HA-T-AGM protein. (**i**) Size distribution profiling, and (**ii**) Apparent zeta potential distributions of the HA-T-AGM protein. (**iii**) Statistical analysis and quantifications of the size diameters and zeta potentials. Here, different colors in the graph represent three independent experiments.

**Figure 4 vaccines-11-00780-f004:**
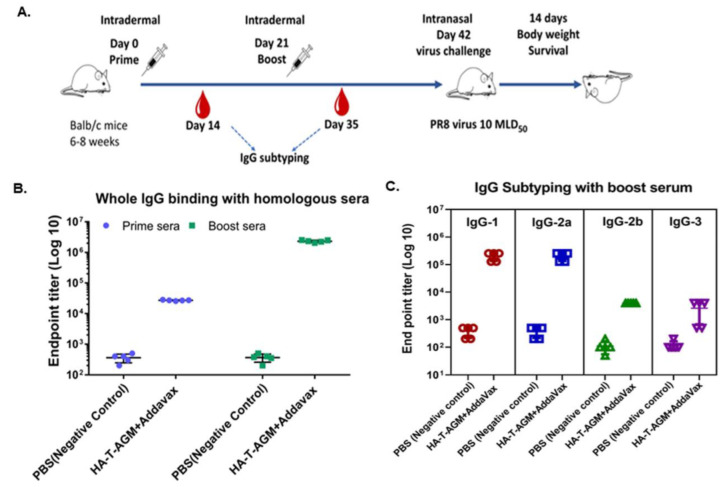
Immune response assessment after prime-boost of HA-T-AGM in mice. (**A**): Schematic representation of antigen administration in BALB/c mice (*n* = 5) for each test group and *n* = 5 for naïve control group). The black needle shows the dosage time point, while the red blood drop indicates the time of blood collection. (**B**): IgG whole (H+L) binding with boost sera using homologous protein and mice group sera. (**C**): IgG subclass IgG1, IgG 2a, IgG 2b, and IgG3 identification from mouse sera as indicated by ELISA using anti-mouse IgG subtype secondary antibody tagged with HRP. Values plotted are the geometric mean titers mean ± S.E. of triplicate wells.

**Figure 5 vaccines-11-00780-f005:**
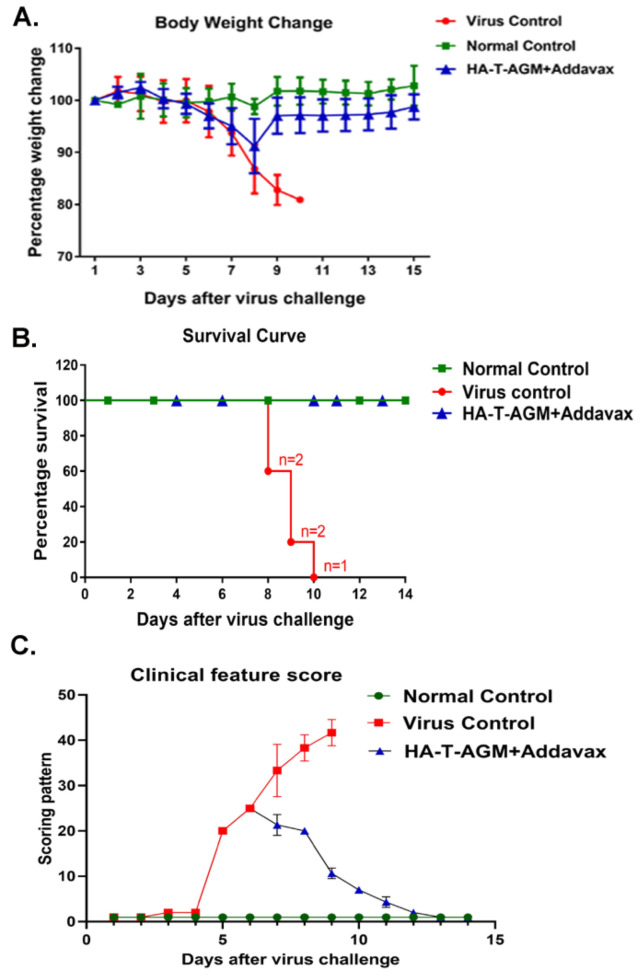
Protection by HA-T-AGM antigen against virus challenge. (**A**): Weight change in immunized mice followed for 14 days post virus challenge (Inf A/Puerto Rico/8/34 (PR8)), and comparison made to the immunized vs. non-immunized (virus control) group along with normal naïve control mice group. (**B**): Survival curve showing the number of dead mice after virus exposure with days passing. (**C**): The animals were scored from day 1 to day 14 after the virus challenge. The scoring scheme followed from 1–10 where 1 shows NAD (No abnormality), 2 was D (Dull), 3 was MP (Mild piloerection), 4 was PE (Piloerection), 5 was RFD (Reduction in food intake), 6 was reduced movement, huddling (RMH) 7 was H (Hunched), 8 was S (Shivering), 9 was SS (Severe shivering) and 10 represents HLI (Hind limb injury).

**Figure 6 vaccines-11-00780-f006:**
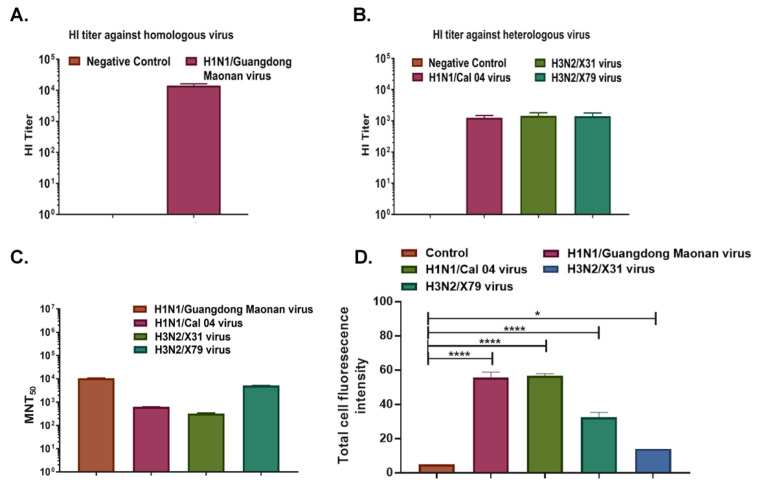
A broad range of cross-protection by HA-T-AGM boost sera against virus subtypes. (**A**): HI titer of Anti-HA-T-AGM vaccinated sera against the homologous strain of the virus (Inf A/H1N1/Guangdong Maonan) (**B**): HI titer of Anti-HA-T-AGM vaccinated sera against heterologous strains of the virus (Inf A/H1N1/Cal 04; Inf A/H3N2/X-31; Inf A/H3N2/X-79) (**C**): Microneutralization Assay showing MNT_50_ titer of HA-T-AGM boost sera against different virus subtypes showing cross-protection. (**D**): Influenza A/Guangdong HA-T-AGM anti-sera cross-reactivity with other Influenza viruses as shown with Immunofluorescence assay. The data represented here is mean with SE. The normality of the samples was assessed with the Shapiro-Wilk normality test; statistical analysis was then performed using Student’s *t*-test. * *p* < 0.05 and **** *p* < 0.0001.

**Figure 7 vaccines-11-00780-f007:**
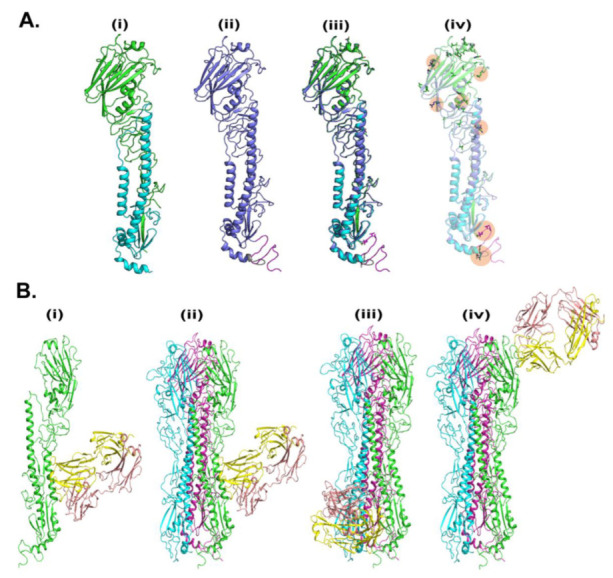
In-silico analysis of HA-T-AGM. (**A**). Molecular modeling of HA-T-AGM protein and structural comparison with reported structures: (**i**) Crystal structure of pdb-id 4LVX, including chain A (green) and chain b (cyan) (**ii**). Model of HA-T-AGM (blue) (**iii**) The overlay structure between crystal and model and (**iv**) The overlay structure (in transparent), to show the changes in amino acids (in licorice). (**B**). Molecular Docking of HA-T-AGM with reported mAb: (**i**) monomer chain (in green) with the most likely pose of mAb (pdb-id 3SDY) (yellow and magenta) (**ii**). Trimer Model of HA in protomer 1 (green), protomer 2 (cyan), and protomer3 (purple) highlighted the most likely pose is consistent in trimer as well (**iii**) 2nd pose of mAb in trimer and (**iv**) 3rd pose of mAb in the trimer.

## Data Availability

The original contributions presented in the study are included in the article/Appendix A, further inquiries can be directed to the corresponding author.

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
