# Peer review of "Intradermal Immunization of Soluble Influenza HA Derived from a Lethal Virus Induces High Magnitude and Breadth of Antibody Responses and Provides Complete Protection In Vivo"

_vaccines, 2023, doi:10.3390/vaccines11040780_

Round 1
Reviewer 1 Report
In the paper of Raj et al., the authors show the design, expression and purification of a recombinant soluble Hemagglutinin glycoprotein. The protein was find to be highly stable, form trimers and oligomers and elicit robust humoral antibody responses after immunization of mice. Antibody induced also protected immunized mice against the lethal virus challenge. Therefore, this immunogen could be considered to be used for the development of subunit protein vaccine. The research topic is interesting, the whole manuscript is well structured. The presented results in the manuscripts are technically sound and logically presented. However, some typo and punctuation mistakes should be corrected before publishing. Also the quality of the figures could be improved, they looked a little bit outstretched.
Author Response
We would like to thank the reviewer for their careful evaluation and insightful comments that have helped us to improve the current revised version of the manuscript. We have addressed all the comments made by the reviewer to improve the clarity of the manuscript. We agree with most of their propositions and updated our manuscript accordingly.
Please see the attachment

Reviewer 2 Report
In this study, authors expressed the ecto-domain (not full length as stated in the manuscript) of hemagglutinin (HA) of A/Guangdong-Maonan/SWL1536/2019 (H1N1), which is the recommended northern hemisphere egg-based flu vaccine strain in year 2020-2021. Actually, this H1 HA has the same protein sequences as 2020-2021 cell- or recombinant based flu vaccine strain A/Hawaii/2570/2019(H1N1). The manuscript missed this critical point. So the H1 HA was already used in recombinant protein based flu vaccines such as commercialized Flublok. However, the different expression system were used: mammalian cell Expi 293F was used in this study, while baculovirus insect cell was used to produce Flublok.
It is interesting to see this soluble ecto-domain H1 HA protein from mammalian cell system, and the protein seems immunogenic in mice immunization study. The immunized mice were protected from the PR8 H1N1 virus challenge, and the sera from the immunized mice showed good HI titer and neutralization titer against the original virus of H1N1/Guangdong-Maonan. However, it is really surprising to see the H1 immunized mice sera has really High HI titer crossing HA group to H3 viruses, and microneutralization experiment showed the immunized mice sera could neutralize with similar titer against H1N1/Guangdong-Maonan and H3N2/X79 virues (Figure 6C), which is totally unexpected. It is well documented that H1 HA immunization can hardly cross over to H3 HA, which is why there are H1N1 and H3N2 as well as flu B viral strains in annual flu vaccines. If the result was true without any human error or contamination, prophylactic and therapeutic experiments of the mice sera should be carried out to H1N1 viruses, H3N2 viruses even flu B viruses. It will be a really high profile research work if authors can prove the H1 HA immunization can cross protect H3N2 viruses and/or even flu B viruses.
CR8020 is a group-2 HA specific antibody, which can not bind to H1 HA. It is not understandable why CR8020 was chosen to dock with the H1 HA in this study. The authors have to prove there is real binding between CR8020 and this H1 HA.
Author Response
We are grateful for the in-depth review of our manuscript and critical comments to improve the manuscript. Please see the attachement.

Reviewer 3 Report
Overall, this is a decent paper describing a novel influenza vaccine. The manuscript could benefit from English language editing beyond the scope of this reviewer.
Major comments:
Why did the authors choose to perform the challenge with a different strain of influenza than the one used to derive the vaccine? There is also confusion regarding comments about challenging with a different strain than what is described in the methods and results. This needs to be clarified and resolved.
The animal numbers here are too low to qualify as anything other than a pilot proof of concept study and this should be described in the discussion as a major study limitation.
Minor comments:
Line 27: Change “have” to “are”
Line 51: remove “however”
Lines 54-57: this is an awkward sentence and should be reworded for clarity.
Line 95: Remove “high”
Line 99: Change to “Our results suggest that the stable …”
Line 168-169: Remove “from RBCs separate”- the sentence should end after “sera.”
Lines 269-271: This strain was not used in the present study, so if this was an additional experiment, it should be added to the methods and if not, this line should be removed or the appropriate work cited.
Lines 281-283: This is a clumsy sentence and I’m not sure if the authors are referring to the A/Michigan strain or some other strain? Please clarify and consider starting the sentence with: “The A/Michigan/45/2015 strain did not lead to a pandemic but its occurrence in humans and pigs suggests it could be a future threat. “
Line 392: Change to “intradermal”
Line 418: Change “viruses” to “virus”
Lines 419 and 420: remove “and symptoms”- we can only assess clinical signs in animals.
Line 421: Emancipated is not appropriate in this context, do the authors mean emaciated? If so, emaciation is a very advanced clinical sign and any animals allowed to reach that state should have been removed from the study earlier for ethical reasons.
Line 448: Does “mice sera protected against tested homologous virus” mean that the serum antibodies neutralized virus? If so, rewrite accordingly- as written, it sounds like this is a passive transfer of immunity study, which it was not.
Line 448 and elsewhere: Change “mice sera” to “mouse sera” in every instance
Line 512: remove “dream”
Lines 528-529: The A/Guangdong-Maonan virus wasn’t used in the study described herein
Line 551: Remove “folds”
Line 552: Change “binding” to “bind”
Line 563: Define the strain here, do not just say “lethal strain”
Author Response
We are grateful for the in-depth review of our manuscript and critical comments to improve the manuscript. We have incorporated the changes as suggested by the reviewer. Please see the attachment.

Reviewer 4 Report
By the immunity induced by intradermal inoculation of recombinant soluble trimeric HA protein, which was derived from H1N1 subtype influenza virus, mice were completely protected from lethal H1N1 virus infection. Surprisingly, the antibodies induced showed neutralizing activity against the H3N2 subtype influenza virus as well as HI activity against type B influenza virus. In the present study, Raj et al showed the potential for a broad-spectrum influenza vaccine that is effective regardless of subtype or type was demonstrated. The present study was properly conducted and contains important information.
I just have the following questions and comments:
At line 45-46, were the H17N10 and H18N11 viruses isolated? Or was the gene detected?
At line 81, the term “pdm-9” should be “pdm09”.
Incorrect notation of subtypes in virus strain names at line 105-106; A/Guangdong-Maonan/SWL1536/2019 (H1N1)
At 198-200, is the determination method for HI test correct? Is it not necessary to tilt the plate to determine the difference between complete and incomplete inhibition?
At line 226-227, abbreviations for virus names (Guangdong, X-31, X-79, and Cal09) should be defined at the first appearance.
It is one of the most important findings in this study that the antibodies induced by intradermal inoculation of soluble trimeric H1 protein showed neutralizing activity against the H3N2 virus and HI activity against influenza B virus. Therefore, the authors should give due consideration to the reasons for such an outcome. The second paragraph from the end is a repetitive description of the results and does not provide the expected discussion. If special effects due to adjuvants are inferred, then adjuvants should be mentioned in the discussion.
Author Response
We would like to thank the reviewer for their careful evaluation and insightful comments that have really helped us to improve the current revised version of the manuscript. Please see the attachment.

Round 2
Reviewer 2 Report
The revised manuscript still did not address the concern from previous comments. It was noticed the authors plan to do more experiments to solve the problems. The manuscript can not be accepted until further experimental results obtained.
Author Response
We would like to clarify that we did not plan to do more experiments to solve any problem. The HA-T-AGM looks like a promising vaccine candidate, hence we plan to take it forward, and to make a commercial product it needs to be tagless. Hence, we plan to make a tagless construct. As the reviewer suggests doing a passive immunization study which is a good suggestion, we will incorporate this in our future studies after the release of project funds.
We explained the funding issue for which one of the experiments with CR6261 and CR8020 mAbs cannot be done currently and we will do in the future which will take some time.
Regards